# Lifestyle Prescription for Depression with a Focus on Nature Exposure and Screen Time: A Narrative Review

**DOI:** 10.3390/ijerph19095094

**Published:** 2022-04-22

**Authors:** Vicent Balanzá-Martínez, Jose Cervera-Martínez

**Affiliations:** 1Centro de Investigación Biomédica en Red de Salud Mental (CIBERSAM), 28029 Madrid, Spain; 2Teaching Unit of Psychiatry and Psychological Medicine, Department of Medicine, University of Valencia, 46010 Valencia, Spain; 3Department of Medicine, University of Valencia, 46010 Valencia, Spain; joecerve@hotmail.com; 4Hospital de Denia-Marina Salud, 03700 Denia, Spain

**Keywords:** depression, nature exposure, screen time, lifestyle

## Abstract

Recent lifestyles changes have favored increased time in contact with screens and a parallel reduction in contact with natural environments. There is growing awareness that nature exposure and screen time are related to depression. So far, the roles of how these environmental lifestyles affect depressive symptoms and disorders have not been reviewed simultaneously. The aim of this review was to gather the literature regarding the role of nature exposure and screen time in depression. An emphasis was made on clinical samples of patients with well-defined depression and the different methodological approaches used in the field. A second goal was to suggest an agenda for clinical practice and research. Studies were included if they assessed depressive symptoms in patients with a clinical diagnosis of depression. An overview of the published literature was conducted using three scientific databases up to December 2021. Several interventions involving nature exposure have shown positive effects on depressive symptoms and mood-related measures. The most consistent finding suggests that walks in natural environments may decrease depressive symptoms in patients with clinical depression. Less researched interventions, such as psychotherapy delivered in a forest or access to natural environments via virtual reality, may also be effective. In contrast, fewer observational studies and no experimental research on screen time have been conducted in patients with clinical depression. Thus, recommendations for practice and research are also discussed. Scarce research, diverse interventions, and several methodological shortcomings prevent us from drawing conclusions in this area. More high-quality experimental research is needed to establish interventions with proven efficacy in clinical depression. At this stage, it is too early to formulate practice guidelines and advise the prescription of these lifestyles to individuals with depression. The present findings may serve as a basis to develop strategies based on nature exposure and screen time targeting clinical depression.

## 1. Introduction

Depression is a prevalent mood disorder. It is estimated that more than 300 million people worldwide suffer from depression [1]. Depression is also a leading cause of disability and a major contributor to the overall global burden of disease [2]. Currently, first-line treatments for depression are standalone or combined psychopharmacology and psychological therapy. However, only a subset of patients achieved clinical remission. Given the unmet needs and potential side effects of these treatments, other strategies have been researched to improve clinical outcomes in depression [3]. Lifestyle-based interventions represent one such approach.

Physical activity, diet/nutrition, and restorative sleep are the three basic pillars of lifestyle. Currently, lifestyle is considered a multidimensional construct, which integrates other behaviors, such as stress management, social support, substance use, and outdoor activities [4].

Compared to the general population, patients with depression show unhealthier lifestyles, including poorer sleep quality and dietary patterns, lower levels of physical activity, smoking, and substance/alcohol abuse [5]. Notably, all these lifestyle factors associated with depression are modifiable [6]. Growing evidence supports that lifestyle behaviors play a role in the treatment and even prevention of several psychiatric disorders, including depression [7]. Lifestyle changes aimed to manage mental disorders are the foundations of the nascent field of Lifestyle Psychiatry [8].

In recent decades, the global changes spurred by urbanization have resulted in remarkable shifts from traditional lifestyles and contact with nature [9]. Together, growing technological developments have favored an increase in time spent in contact with screen-based devices and a parallel reduction in time spent in nature. According to a recent scoping review [10], excessive screen time seems to be associated with negative psychological outcomes, including mental health and cognitive function, whereas time in contact with nature seems to be associated with favorable psychological outcomes [10]. Nevertheless, nature exposure and screen time are among the least researched lifestyles in depression so far. To the best of our knowledge, the relationship between these environmental lifestyles and depressive symptoms/disorders has not been reviewed simultaneously.

A growing number of reviews and meta-analyses supports the link between nature exposure and depression [11,12,13,14,15,16]. For instance, short-term exposure to natural environments has shown positive but small effects on depressive mood [16]. However, this evidence mostly relies on primary studies examining either healthy subjects or patients with depressive symptoms but not a primary diagnosis of clinical depression. Experimental studies focusing on clinical depression are far fewer and have not been synthesized so far. Moreover, previous reviews adopted a restricted scope regarding nature exposure. For instance, they focused on one type of natural environment, such as forests [14,15], or a specific intervention, such as nature walks [13] and gardening [14]. Interestingly, no previous literature review was restricted to patients with clinical depression. Taken together, an overall understanding of how nature exposure impact depressive symptoms in patients with clinical depression represents a gap in the literature.

This review is intended to address that gap by undertaking two complementary approaches. On the one hand, we only selected studies that included populations with clinical depression. On the other hand, we adopt a wider perspective on these topics. For instance, here we include all types of natural environments and nature-based interventions (NBIs). Although a consensus definition is lacking, NBIs are defined here as ‘planned, intentional activities to promote individuals’ optimal functioning, health and well-being or to enable restoration and recovery through exposure to or interaction with either authentic or technological nature’ [17]. Screen time (ST) is defined as the viewing or use of any technology with a screen, including television, computers, and mobile devices such as smartphones and tablets [18].

The aim of this review was to gather the literature regarding the role of nature exposure and screen time in depression. An emphasis was made on clinical samples of patients with well-defined depression and the different methodological approaches used in the field. A second goal was to suggest an agenda for clinical practice and research.

## 2. Methods

This paper was planned as a narrative overview of the scientific literature, including a critical discussion of retrieved evidence [19]. This review was intended to provide a reference for wider reading instead of a systematic review [20]. Therefore, no systematic literature search was attempted.

To conduct this overview, electronic databases (PubMed/MEDLINE, Science Direct and Google Scholar) were searched up to 15 December 2021. Search terms related to depression (e.g., ‘major depression’, ‘unipolar depression’, and ‘depressive disorder’), nature (e.g., ‘natural environment’, ‘nature exposure’, nature contact’, ‘green space’, ‘blue space’, ‘nature-based intervention’), and screen time (e.g., ‘screen time’, ‘media use’, ‘screen use’, ‘screen media’, ‘social media use’) were used. The electronic search was supplemented by hand-searching of reference lists of reviews and meta-analyses to identify additional articles. Moreover, a snowballing method was used, searching the references of relevant articles to identify further suitable publications [20]. We considered only published articles and not grey literature. We included human observational and experimental studies on the relationship between depression and nature exposure or screen time. Studies were included if they assessed depressive symptoms in patients with a clinical diagnosis of depression, regardless of the sample included other conditions (termed ‘mixed samples’ in the manuscript). We prioritized studies with clinical samples composed of patients diagnosed with standardized criteria of depression, e.g., DSM (Diagnostic and Statistical Manual of Mental Disorders) and ICD (International Classification of Disease). Case reports, opinion articles, and articles not published in English were excluded. Studies about other environmental exposures, such as light at night and air pollution, were purposely excluded from this review.

## 3. Results

### 3.1. Nature Exposure and Depression

Several observational and intervention studies assessing the relationship between natural environments/NBIs and depression have been published to date.

#### 3.1.1. Observational Studies

Three observational studies were identified, although none recruited a clinical sample restricted to participants with clinical depression [21,22,23]. In a web-based, cross-sectional survey, Ower et al. [21] compared levels of physical activity (PA) developed indoors, outdoors, or in an alpine environment in healthy controls and patients with psychosomatic disorders, mostly somatoform disorder, and major depressive disorder (MDD). In addition, possible mediating variables such as an individual’s resilience and quality of life were analyzed. Only PA carried out in an alpine environment was associated with greater resilience. Furthermore, the effect of PA carried out in alpine environments on quality of life was partially mediated by resilience in both patients and controls. In a retrospective cohort study with a matched sample, Währborg et al. [22] assessed healthcare consumption in patients with either ICD-10 depression or reactions to severe stress that used to participate in a multimodal nature-assisted rehabilitation program. This 12-week program combined horticultural activities with traditional rehabilitation conducted in a green setting, e.g., a garden designed for this purpose. A reduction in the number of visits to primary care was observed, as well as the mean hospital stay for both psychiatric and somatic causes. An innovative approach was adopted in the third study [23]. Ecological Momentary Assessment (EMA) was used to examine the association between daily encounters with nature or daylight and affect-related indicators. Several times per day, participants were asked to report the number of natural elements they were exposed to and their affective state, as measured with 12 mood items derived from the Thayer Activation-Deactivation Checklist and University of Wales Institute of Science and Technology Mood Adjective Checklist. Nature exposure and daily light were associated with significant benefits on the affective state, especially in patients with higher levels of depressive symptoms. However, the standardized diagnostic criteria of depression, e.g., ICD or DSM, were not used to recruit participants in this study.

#### 3.1.2. Intervention Studies

Several intervention studies were identified [24,25,26,27,28,29,30,31,32,33,34,35,36,37,38,39]. Of them, a minority included patients with MDD only [24,25,26,27,28,29], and the majority [21,30,31,32,33,34,35,37,38,39] recruited mixed samples including patients with depressive disorders, although separate results for that group were not reported.

##### Intervention Studies with Samples of Clinical Depression Only

A few intervention studies have examined the effects of nature contact/exposure on clinical outcomes in patients with clinical depression only [24,25,26,27,28,29]. The main characteristics of these studies appear in Table 1.

In the study by Kim et al. [25], outpatients with DSM-IV-TR depression of at least moderate severity were assigned to three interventions over four weeks: cognitive behavioral therapy (CBT)-based psychotherapy delivered in a forest environment, the same psychotherapy conducted in the hospital and usual management. Compared with the other two interventions, the forest activity significantly decreased depressive symptoms (HRSD baseline and endpoint total scores: 15.54 to 8.04 vs. 15.79 to 11.58 in the hospital group and 16.90 to 14.85 in the control group) and significantly improved rates of clinical remission (61% vs. 21% in the hospital group and 5% in the control group). Interestingly, only the forest program was associated with reduced concentrations of salivary cortisol, which is a biomarker of stress [25].

Building upon previous studies [26,30], Korpela et al. [28] added nature walks to a ‘Coping with Depression’ program [40] in 13 patients with ICD-10 depression (see more details in Table 1). Nature walks are ‘walks on a trail in a natural setting such as forests, woodlands, or parks that can take any time from 15 min to a few hours’ [12]. A significant decrease in depressive symptoms (Wilcoxon r = −0.49) and a significant increase in mental well-being (r = 0.46) were observed at the end of the 8-week intervention and at three-month follow-up. Nature walks, but not indoor sessions, produced restorative experiences that were found to mediate these changes.

A cross-design study analyzed the role of attention and rumination in 20 patients with DSM-IV moderate to severe MDD after taking a 50 min walk in either natural or urban environments [26]. Prior to the intervention, patients with MDD were asked to think about an unresolved negative autobiographical event to prime rumination, which is a common symptom of depression. Before and after each condition, mood and working memory were assessed with PANAS and the digit span backwards, respectively. Compared to the urban condition, taking a walk in a botanical garden resulted in greater improvements in positive affect (η_p_^2^ = 0.29) and cognition (working memory, η_p_^2^ = 0.53). However, these results were not correlated, suggesting that separate mechanisms underly the observed benefits.

Frühauf et al. [27] compared the effects of indoor and outdoor PA in a sample of 14 inpatients with mild to moderate depression, as assessed by a clinician. The outdoor PA consisted of Nordic walking in a green natural environment, while the indoor PA consisted of cycling inside a gym. A sedentary condition in which patients sat and read materials and board games was included to control the effects of group activity. All patients participated in three interventions comprising 60 min group sessions conducted on different days. Affective valence and perceived activation were evaluated with the Feeling Scale and Felt Arousal Scale, respectively. Moreover, mood states were assessed with the Mood Survey Scale (MSS), which includes measures of anger, excitement, activation, fatigue, and depression, among others. Outdoor PA was found to be more beneficial than indoor PA by generating significant changes in patients’ excitement (Cohen’s d = 1.45) and activation (d = 2.04). The benefits in perceived activation were observed after 45 min of PA in the outdoor. Additionally, improvement in patients’ levels of anger, fatigue, and depression (d = 1.08) were found at the trend level.

Another study focused on elder patients with mild to moderate depression. McCaffrey [24] developed a qualitative study based on focus group interviews. Depression was either diagnosed by a healthcare provider or based on patients’ self-report. Participants were randomly assigned to three intervention groups: walking alone or guided walks in a garden setting and art therapy as a control group. Healing gardens, in both guided and non-guided groups, were found to be helpful to relieve patients’ depression and improve their mood.

Finally, Szczepanska-Gieracha and colleagues [29] evaluated the efficacy of accessing nature using virtual reality (VR) in elder women with depression who had not responded to previous treatment. All patients (n = 25) participated in a psychoeducational program and general fitness training and those randomized to the intervention group (n = 13) completed eight sessions of VR therapy. By means of a head-mounted display, patients accessed a total immersive experience in an interactive therapeutic garden, which was modified as the sessions progressed, and mood changed. A significant decrease in depressive symptoms as assessed with the Geriatric Depression Scale (GDS-30) was observed post intervention (12.27 vs. 8.27, *p* = 0.001) and was maintained at 2-week follow-up (8.27 vs. 7.27, *p* = 0.88).

##### Intervention Studies with Mixed Samples

Roe and Aspinall [30] conducted one of the first quasi-experimental studies that compared walks in urban and rural settings. Two groups of participants were recruited—a mixed sample of patients attending mental health services (poor mental health group; n = 40) and adults from existing walking groups (good mental health group; n = 83). Walks in rural areas were significantly correlated with positive changes in mood in both groups but were greater in those with poor mental health. Interestingly, significant benefits of urban walks were observed only in the latter group, for whom both urban and rural walks were associated with positive changes on hedonic tone and stress, assessed with the Mood Adjective Checklist.

Barton et al. [31] developed a pre-post intervention study with a mixed clinical sample, including patients (n = 53) with DSM-IV-TR diagnoses of mood, anxiety, and psychotic and substance-related disorders. The green exercise program consisted of six guided walks that took place once a week in local public green spaces, such as country parks and natural reserves. In order to control the effects of the social interaction and PA ingredients of the green exercise program, two control groups—a social club group and a swimming group—were also included. Self-esteem was assessed by Rosenberg Self-Esteem Scale (RSE), and mood was assessed by the 30-item short-form version of the Profile of Mood States (POMS) questionnaire. After participating in a single session, an improvement in both self-esteem and mood was observed in all groups, which was greater in the green exercise group (self-esteem: 23.6 vs. 21.0, *p* < 0.0001; overall mood: 154.3 vs. 143.1, *p* < 0.0001). There was evidence for a dose–response relationship, with increasing benefits after 6 weeks of treatment.

Bielinis et al. [32] tested the efficacy of a forest therapy program for hospitalized patients with ICD-10 psychotic and affective disorders, including depression. Forest therapy or forest bathing (*shinrin yoku* in Japanese) refers to visiting a forest or engaging in various therapeutic activities in a forest environment to improve one’s health and wellbeing [41,42]. In a quasi-experimental design study (n = 50), without a control group, they assessed the effects of walking, stretching, or watching landscapes in a forest environment on anxiety and mood state. In the affective disorders group, a single intervention of less than two hours was associated with improvements in variables such as tension–anxiety (Cohen’s d = 1.77), depression–rejection (d = 2.52), fatigue (d = 1.27), confusion (d = 3.46), and vigor (d = 1.71), as measured with the State-Trait Anxiety Inventory-State (STAI-S), and the Profile of Mood States (POMS).

Vujcic et al. [33] explored the benefits of a horticulture therapy program for a mixed sample of 30 patients with ICD-defined adjustment disorder and a reaction to severe stress, anxiety, or depression, who attended a day hospital. Horticultural therapy is a process through which garden-related activities, interaction with plants and closeness to nature are used as a rehabilitative strategy [43]. After 12 sessions, developed in a botanical garden, significant benefits were observed in depression, stress, and anxiety and were assessed with the Depression Anxiety Stress Scale (DASS21).

Moreover, Triguero-Mas et al. [34] differentiated natural outdoor environments (NOE) into green and blue spaces, which were compared to the ‘built environment’ in an urban district. Using a randomized, case-crossover design, they analyzed the effects of exposure to PA in the three environments on psychological (measured as Total Mood Disturbance, TMD) and physiological variables, including salivary cortisol levels, blood pressure, and heart rate variability. Participants (n = 26) had previously scored below the 50th percentile on the Mental Health Inventory scale (MHI-5). Exposure to a green space (a natural park) was associated with significant reductions in mood disturbance (−4.78 points in TMD score) and salivary cortisol (−0.21 log nmol/L), while exposure to a blue space (a beach) improved mood (−4.53 points in TMD score) and heart rate indicators. Interestingly, PA and self-perceived restoration experience were found to partially mediate the association between NOE exposure and mood.

Maund et al. [35] conducted a 6-week, multimodal intervention in a natural setting, including bird watching, free or guided walks, and canoeing. A mixed clinical sample of 16 patients with depression and/or anxiety was recruited. Significant improvements in anxiety (Generalized Anxiety Disorder-7), stress (Perceived Stress Scale), and emotional wellbeing (Positive and Negative Affect Schedule and Warwick Edinburgh Mental Wellbeing Scale) were observed after the program. Significant changes in the latter scale were assumed to represent improved depressive symptoms (37 vs. 41 points; *p* < 0.009). Moreover, in a qualitative analysis, patients highlighted a reduction in social isolation, an increase in confidence to be in nature, and an improvement in the management of PA.

In a laboratory investigation, Hüfner et al. [36] exposed healthy controls and patients with mental disorders, such as somatoform, depressive, and anxiety disorders, to visual stimuli from alpine environments vs. neutral stimuli. A positive effect of alpine vs. neutral visual stimuli, assessed by Self-Assessment Manikin, was observed on emotional analytics for both groups.

Suicidal behavior, which is usually associated with depression, has also been targeted by NBI. Sturm et al. [37] carried out a randomized crossover trial with a small sample (n = 20) of high-risk suicide patients with depression and other psychiatric disorders. The intervention, consisting of a 9-week, mountain hiking program, was effective in reducing hopelessness (assessed by Beck Hopelessness Scale; d = −1.4, *p* < 0.0001), depression (assessed by Beck Depression Inventory; d = −1.38, *p* < 0.0001), and suicide ideation within the hiking phase (assessed by Beck Scale of Suicide Ideation; d = −0.79, *p* = 0.005). Despite their severe clinical state, patients were found to be sufficiently motivated to participate in the program.

Regarding adolescents, Bowen et al. [38] developed a 10-week program based on Wilderness Adventure Therapy (WAT) [44] for outpatients with mixed psychiatric diagnoses (n = 36), mostly depression and conduct disorder. The WAT model emphasizes the development of social-emotional competencies and coping skills through group-based adventure experiences that are facilitated by a psychologist. In this case, activities included bushwalking, abseiling, cross country skiing, or white-water rafting. A post-intervention improvement of self-esteem (Hedge’s g = 0.26) and psychological resilience (g = 0.49) was observed, as well as a reduction in depressive symptoms (g = 0.46), especially in participants with worse baseline scores. At three-month follow-up, the improvement in depressive outcomes was maintained.

Finally, Gonzalez et al. [39] developed a study with a single-group of 28 patients with DSM-IV defined MDD, dysthymia, or depressive phase of bipolar disorder type II, which participated in a 12-week therapeutic horticulture program. Compared to baseline, mean scores on the Beck Depression Inventory (BDI) significantly decreased 4.5 points at the end of the intervention (*p* = 0.002) and were maintained at the three-month follow-up. The intervention was also associated with an improvement in attention, as measured with the Attentional Function Index (AFI) scores.

#### 3.1.3. Potential Causal Mechanisms

The positive effects of nature exposure on depressive symptoms likely result from the interaction of multiple mechanisms. Two major theories have been invoked to explain these benefits: attention restoration theory [45] and stress reduction theory [46]. Attention restoration theory (ART) proposes that time spent in natural environments provides the experience of a ‘soft fascination’ that allows the ability to pay attention without effort. ART has been suggested to explain the results of some of the reviewed studies [33]. Other clinical studies highlighted the importance of attention [39] or rumination [26] as factors involved in mood improvement after exposure to natural environments.

Stress reduction theory (SRT) suggests that natural settings evoke an ‘automatic positive affective response’, which reduces stress and physiological activation [16,47]. It has been suggested that stress relief after nature exposure can regulate the immunological response [47]. Direct physiological mechanisms, such as exposure to microorganisms and volatile organic compounds and microorganisms, would also enhance immune functioning [48].

Another major pathway linking nature and depression involves enabling of health-promoting behaviors, such as social contacts and PA [47]. According to some controlled studies of patients with depression, clinical improvement cannot be explained only by the effects of PA or social interaction [27,31]. At the brain level, it has been suggested that viewing images of nature can improve mood by reducing the activity of the orbitofrontal cortex, which is hyperactive in patients with depression at rest [49]. Moreover, there is evidence that nature exposure can reduce the activity of the subgenual prefrontal cortex and rumination in healthy subjects [50].

### 3.2. Screen Time and Depression

The growing development and use of digital technology has accelerated people’s exposure to screen-based devices [51]. A positive association between screen time (ST) and mental health problems, including depressive and anxiety symptoms, has been reported [52,53]. Most of this research has been conducted in young people from the general population (see below), whereas fewer studies focused on adults. Thus, results are presented according to age group.

#### 3.2.1. Screen Time and Depression in Adults

Three large, nationally representative samples of adults from the general population have examined the link between ST and depressive symptoms/disorders across cultures and societies [18,54,55]. A large population-based, cross-sectional study of US adults found a significant association between ST and moderate or severe levels of depression after controlling for potential covariates [54]. Moreover, individuals who spent more than six hours per day watching TV and using computers had higher odds of developing depressive symptoms. In another study, Yu et al. [18] analyzed time spent by Chinese adults in mentally passive (e.g., watching TV) and mentally active (e.g., using computer and mobile devices) activities separately. Watching TV was associated with increased depressive symptoms and the reverse was found for more mentally active modalities of ST. Finally, Werneck et al. [55] compared the unhealthy lifestyles of adults with severe mental illnesses with those of the general population in Brazil. MDD was associated with increased TV watching, tobacco use, and consumption of sweets and soft drinks. Moreover, an analysis of more than 84,000 adults from the UK Biobank [56] found that multiple lifestyle factors were associated with depressed mood. Increased ST as well as poor sleep and dietary pattern were considered partly implicated in the development and aggravation of depressed mood.

Social media use is a topic related with ST. In a nationally representative sample of young adults from the United States, social media use was significantly associated with increased odds for depression [57]. Strong and linear associations were found for all three measures of social media use—total time per day spent on social media, visits per week, and a global frequency score. In a national cohort study of young adults, among individuals who were initially not depressed, baseline social media use was independently associated with the development of depression over the following six months [58]. However, baseline depression was not associated with an increase in social media use at follow-up in that study.

Fewer studies have assessed adult patients with clinical depression. Two observational studies with clinical samples have been published recently [59,60]. Tønning et al. [59] assessed 74 adult patients with ICD-10 unipolar depressive disorder over six months following discharge from psychiatric hospitalization. The daily number of times the screen was switched on and the screen-on duration were used as proxy measures for smartphone usage. Higher levels of depressive symptoms, measured with the Hamilton Rating Scale for Depression (HRSD-17), were associated with fewer times the screen was switched on. These findings support that smartphone-based self-monitoring is feasible and associated with clinical ratings in patients with depression. Another recent study [60] examined the use of social media in outpatients with unipolar depression and bipolar disorder. After controlling for age and sex, both clinical groups and the healthy control group spent a similar amount of time on social media. The use of social media followed a mood-congruent pattern, with decreased and more passive use during depressive episodes. Interestingly, more participants reported detrimental than beneficial effects on mood symptoms during depressive episodes [60]. Finally, to the knowledge of authors, no experimental research to date has involved ST in patients with depression.

#### 3.2.2. Screen Time and Depression in Young Populations

Young people, e.g., children and adolescents, are especially exposed to ST and adolescence is a period of high risk for the onset of depression [61]. In adolescents, excessive ST has been associated with mental health outcomes such as psychological problems and greater risk for depression or anxiety [51]. Overall, results regarding the relationship between ST and depressive symptoms are inconsistent [53,62,63,64].

The high heterogeneity and the small overall effect size observed suggest that other factors likely moderate the relationship between depressive symptoms and ST in adolescence [65]. The type of screen-based activity seems to be one such moderator. For instance, mentally active activities, such as video gaming, may even have protective associations with depressive symptoms in adolescents [66]. As noted by Kim et al. [67], most studies have not differentiated between active and passive forms of ST. In a representative sample of adolescents aged 12–17 years, past six-month incidence of DSM-IV-TR defined major depressive episode was associated with passive ST but not with active ST [67]. Conversely, in a large Swedish cohort study, self-reported hours spent in passive, sedentary behaviors, including screen use, were not significantly associated with the incidence of MDD over a 13-year follow-up [68]. Still, other studies found positive associations between active ST and depressive and anxiety symptoms in adolescents [69]. Individual characteristics might represent another important moderator. In a large cross-sectional study with children and adolescents, the association between ST and depression varied with screen type and individual characteristics, such as gender and developmental stage [70]. During adolescence, heavy screen media use was more strongly associated with mental health issues, among girls compared to boys, especially for social media and Internet use [71]. On the other hand, sleep and PA are consistently replicated correlates of excessive ST use in adolescents. Indeed, there is growing awareness that increased ST and insufficient sleep and PA are co-dependent and may have synergistic effects to impair mental health [72]. These three lifestyle behaviors should be considered simultaneously when examining their relationship with depressive symptoms in adolescence [67,73].

Moreover, the associations between ST and depression are largely based on the findings of cross-sectional studies. Longitudinal studies suggest that the dynamic interplay between excessive ST and depressive symptoms appears to be complex and bidirectional in nature [74,75]. A recent systematic review of longitudinal studies found small to very small but significant associations between total ST and subsequent depressive symptoms, but not anxiety, in young people [75]. This relationship was stronger than the reverse association between depressive symptoms and subsequent ST.

## 4. Discussion

There is growing research interest in the role of nature and screen time in depressive symptoms/disorders. This overview gathers promising evidence that several NBIs may improve depression outcomes. Specifically, in studies with mixed clinical samples, walks in rural settings and green spaces showed positives changes in mood [26,30], whereas PA in both green and blue spaces improved mood [34]. Moreover, a multimodal program developed in nature improved emotional wellbeing [35] and forest therapy improved depressive symptoms in inpatients with affective disorders [32]. A significant reduction in depressive symptoms was observed after therapeutic horticulture [33,39] and WAT [38]. More consistent findings emerged from the few studies of patients with clinical depression only. Psychotherapy delivered in a forest significantly reduced depressive symptoms and increased rates of clinical remission [25]. Moreover, a virtual reality program was effective to ameliorate symptoms in elder women with treatment-resistant depression [29]. Interestingly, four studies found that walking in natural environments was associated with benefits on depression-related measures [24,27,28,31]. Compared to control interventions, walks in nature decreased depressive symptoms [24,28] improved positive affect [31] and showed a trend to improve mood states [27]. This concurs with meta-analytic evidence that PA conducted in natural settings is associated with greater positive effects on depression compared to indoor exercise [76]. Previous evidence showing the benefits of NBIs for depressive symptoms mostly relied on studies with healthy subjects and clinical populations of patients without a formal diagnosis of depression [12,13,14,15,16,77]. The present review builds upon that literature and expands that knowledge to the few intervention studies comprising patients with clinical depression.

In contrast, the relationship between ST and depression is more equivocal at the time of writing. As reviewed above, very few observational studies and no experimental research on ST have been conducted in patients with clinical depression. Most studies are cross-sectional and have been focused on youth from the general population. The relationship between increased ST and depression may be moderated, at least in part, by screen type, participants’ demographic features, and other lifestyles, such as sleep and PA/sedentary behavior. Based on the available longitudinal studies, the nature of this relationship is likely complex and bidirectional.

### 4.1. Study Shortcomings and Recommendations for Future Research

In this section, the major methodological limitations of the reviewed studies are summarized together with research recommendations to overcome study shortcomings.

Much of the current evidence on the role of NBIs in depression is based on studies with small samples, mixed clinical samples, and heterogeneous designs. Studies with small samples are prone to type II error and may hinder the identification of subgroups of patients that benefit the most. Related to this, recruiting only patients with depression has proven to be difficult [30], which often leads to mixed samples in most experimental studies (see Section 3.1.3).

Study designs are largely variable from observational studies to randomized controlled randomized trials (RCTs). Randomization is not always feasible, ethical, or acceptable in natural environments [78]. Similarly, NBIs cannot be blind because participants and healthcare providers experience the treatment environment [14,25]. Failing to blind the researchers can lead to placebo effects in participants [12]. Moreover, many of the reviewed studies are subject to selection bias because participants are not fully randomized, and assignation to the NBI is based on individual preferences. Self-referred participants might be expected to have an interest in natural environments and NBIs, which may lead to a potential ‘nature-positive’ bias [79]. In sum, RCTs are difficult to apply to NBIs.

Studies are also heterogeneous regarding treatments. Participants received stand-alone NBI or additional interventions, such as antidepressant medication, psychotherapy, and occupational therapy [25,32,34]. In many studies, depressive symptoms were assessed with self-report questionnaires instead of more reliable measures, such as clinician-rated scales.

The control of confounding factors is also difficult in NBIs, either due to the absence of control groups or the difficulty in controlling some covariates. Variables such as time; social interaction; familiarity with locations; pollution or seasonal variations; and sunlight, vitamin D, temperature, humidity, or type of PA have been suggested as potential biases [30,31,34].

Some benefits were observed after a single session of NBI in two studies [31,32]. This concurs with evidence supporting that short-term exposure to nature can improve mood [16]. However, the long-term effects of NBIs on clinical depression are largely unknown. Three of the reviewed studies followed patients up to three months after the intervention [28,38,39] and all found that improvement in depressive symptoms were maintained at follow-up. More longitudinal research is warranted to establish the potential sustainability of these effects.

The quality of risk of bias of the included studies was not assessed in this overview. Overall, the aforementioned shortcomings suggest a low quality of many NBI studies in this area [12,14,16].

Regarding research about depression and *screen time*, the major limitations involve cross-sectional designs and self-assessment of ST and depressive symptoms. Firstly, causal relationships cannot be derived from cross-sectional studies. More longitudinal studies and RCTs are warranted to establish the directionality of the associations observed between ST and depression [64]. Secondly, ST is usually assessed with retrospective self-reports, which may be affected by to recall bias and measurement error. Therefore, a combination of validated objective measures, such as accelerometers, and real-time self-reports, such as EMA, is highly recommended [53,64]. Thirdly, the next generation of studies should use diagnostic psychiatric interviews to examine depressive symptoms instead of self-reported questionnaires.

Finally, why address nature exposure and ST at the same time? These environmental lifestyles can interact at different levels. Growing urbanization and technological developments have favored an increase in ST and a parallel reduction in time spent in nature [9]. It has been suggested that as individuals spend more time with screens, it is likely that they engage less in other activities, such as in-person social interactions, physical activity, and/or time spent in natural environments [9]. Moreover, technology and nature have been suggested to exert opposing influences on human brain and cognitive functions [80]. The possibility exists that nature exposure might buffer psychological impacts of excessive ST. According to a recent scoping review, very few studies have assessed time spent with screens and nature simultaneously and their likely interactive psychological effects in children and adolescents [10]. Further research on this area may pave the way to multimodal lifestyle-based interventions in depression.

### 4.2. Implications for Practice

The importance of lifestyle in the management of depression is gaining momentum. Indeed, for milder cases of depression, recent clinical guidelines recommend addressing lifestyle behaviors before starting conventional treatments [81]. Moreover, several recommendations on diet/nutrition [82,83] and physical activity [84] aimed to prevent and treat depression have been issued.

Health care providers are in a key position to recommend and prescribe *nature* exposure to patients with depression [85]. A replicated finding emerging from studies of patients with clinical depression concerns the benefits of exercise in natural environments for depressive symptoms. This relates to the field of ‘green exercise’. There is growing evidence that being physically active or exercising within a natural environment or green space (‘green exercise’) provides greater mental health benefits than PA or nature contact alone [86]. The efficacy of exercise to prevent and treat depression has been demonstrated [7] and prescribing PA/exercise is endorsed in guidelines [84]. Further high-quality research is needed to confirm the efficacy of green exercise in patients with clinical depression before establishing its prescription in clinical practice. In the future, the potential efficacy of other NBIs, such as therapeutic horticulture and forest therapy, can be also considered regarding specifically clinical depression. Interestingly, the existing literature does not seem to make specific recommendations in terms of exposure time addressed to people with clinical depression. This could be due, among other reasons, to the lack of a standardized method to measure time spent in nature or nature contact [87] as well as the broad diversity of NBIs [88]. In our opinion, the nature-based perspective would be applicable at different levels of clinical practice, from the modification of depressed patients’ lifestyles to the setting in which treatments are developed or the creation of specific protocols for therapeutic programs.

Virtual reality (VR) is a new method of accessing nature with a potential interest for patients with disabling, chronic conditions [89], such as psychiatric disorders. In this regard, a recent meta-analysis showed the efficacy of VR to reduce anxiety levels in patients with a wide range of anxiety disorders [90]. VR was also shown to improve depressive symptoms, although at that time none of the studies focused on patients with MDD [90]. The positive results of a more recent study [29] await replication with independent samples of clinical depression. In healthy subjects, VR has been suggested to be associated with less intense benefits for mood compared to authentic nature contact [91].

Regarding ST, there is limited evidence to guide recommendations on advisable or safe exposure to screens [63]. For instance, the Canadian guidelines for sedentary activity [92] recommend that adolescents engage in no more than two hours of ST per day. The findings of original studies are quite heterogeneous in this regard. For instance, adolescents reporting four or more hours of passive screen time per day were three times more likely to develop a major depressive episode compared to those reporting less than two hours [67]. In other studies, individuals who spent at least two hours per day of ST-based sedentary behavior were more likely to have depression [53]. Further studies are clearly needed to better estimate the dose–response relationship between ST and depression and tentatively establish appropriate time limits for ST [62]. It is important to note that the thresholds at which ST becomes detrimental for mental health may differ according to screen type [93]. Accordingly, both screen modality and the amount of time spent should be considered when developing more tailored recommendations for young people.

Excessive ST has been suggested to displace physical activity, getting adequate sleep, in-person social interactions, and academic activities [10]. It is not our contention that the increased use of screens is without benefits for mental wellbeing and health. Indeed, moderate screen time may provide opportunities to reinforce actual relationships, forge new social interactions, and enhance academic performance [94] Beyond social support, smartphones and other screen-based technologies are currently used to foster physical activity, sports, and aerobic exercise through Youtube videos and online tutorials. *Apps* can be used to improve dietary patterns and stress management. Notably, all these healthy lifestyles are protective behaviors for mental health [7,95].

Given that no experimental research involving ST has been conducted in patients with depression so far, practice recommendations may be extrapolated from indirect evidence. A recent prospective cohort study of adolescents in the UK explored whether replacing any type of ST with exercise at age 14 could reduce emotional distress at age 17 [93]. Only replacing watching TV or using social media with team sports, but not individual exercise, was associated with lower emotional symptoms at follow-up. However, depressive symptoms were not specifically assessed in that study. On the other hand, digital detoxification is one strategy to promote healthy digital habits and minimize the negative impact of ST on health [96]. Conducting RCTs in patients with clinical depression are expected to advance the field.

Taken together, the available evidence is limited to formulate practice guidelines and prescription of these environmental lifestyles to individuals with depression.

### 4.3. Strengths and Limitations of Review

This overview is subject to several limitations. Relevant studies may have been overlooked due to several reasons. First, this was a non-systematic review, which was limited to three databases. Second, owing to the broad and diverse vocabulary used for NBIs [88], significant search terms may have been omitted. Third, only articles written in English were included. Moreover, the risk of bias of included studies was not examined. In narrative reviews such as the present one, the included studies may be biased, relevant literature may have been missed, and the search is not replicable by other authors. However, our study also has some strengths. We adopted a wide approach aimed to gain an overall understanding of how nature exposure impacts the clinical outcomes of patients with depression. To the best of the authors’ knowledge, this is the first time that nature exposure and screen time have been simultaneously reviewed in relation to clinical depression.

## 5. Conclusions

This overview examined the role of nature exposure and screen time in depression, with a special focus on clinical samples of patients with well-defined depression. Scientific knowledge about the role of natural environments and NBIs targeting depression seems to be allocated in different silos. Undertaking a wide perspective can be useful to join the dots in cross-disciplinary fields. There is promising evidence suggesting that nature and screen time are related to clinical depression. Several interventions involving nature exposure have shown positive effects on depressive symptoms and mood-related measures. The most consistent finding suggests that walks in natural environments may decrease symptoms in patients with clinical depression. Less researched interventions, such as psychotherapy delivered in a forest or access to natural environments via virtual reality, may also be effective. In contrast, the relationship with screen time remains more equivocal as very few observational studies and no experimental research have been conducted in patients with clinical depression. Overall, evidence on these topics is scarce, sparse, and presents several limitations. At this stage, drawing firm conclusions is challenging due to the diversity of interventions and methodological designs. An agenda for practice and research is also suggested. More high-quality experimental research is needed to better establish the efficacy of NBIs in clinical depression. The joint study of nature exposure and screen time in depressed individuals can clarify their likely interactive effects. Collectively, it is too early to formulate practice guidelines and advise the prescription of these environmental lifestyles to individuals with depression.

## Figures and Tables

**Table 1 ijerph-19-05094-t001:** Main characteristics of the studies that included samples of patients with clinical depression only.

Authors (Country)	n	Study Design	Evaluation Timing	Intervention Details	Settings	Control Groups	Depression Measure	Other Variables
McCaffrey, 2007 (USA) [24]	60	Qualitative Focus group Randomized	One focus group the last day of participation	Weekly sessions during 6 weeks Group 1 (n = 20): walks alone in a garden setting Group 2 (n = 20): guided walks in a garden setting Group 3 (n = 20): art therapy	Garden	Yes (Art therapy group)	Qualitative information from focus group	None
Kim et al., 2009 (South Korea) [25]	63	Quantitative Quasi-experimental non-randomized study	Weekly	4 cognitive-behavior therapy sessions during 4 weeks (3 h/session) performed in different settings: Group 1 (n = 23): forest Group 2 (n = 19): hospital Group 3 (n = 21): usual outpatient management	Forest garden	Yes (Usual out-patient management)	HRSD MADRS BDI-II	Heart Rate Variability Salivary Cortisol concentration SF-36
Berman et al., 2012 (Canada) [26]	20	Quantitative Randomized and controlled experimental trial	Pre and post-intervention	After a pre-intervention assessment, participants were encouraged to ruminate by instructing them to analyze their emotions about an unresolved negative autobiographical experience. After that, they were randomly assigned to a predefined 50 to 55 min walks alone in nature vs. urban environments.	Park	Yes (Urban walking group)	PANAS	Backwards Digit Span test
Frühauf et al., 2016 (Austria) [27]	14	Quantitative Quasi-experimental non-randomized study	Pre-intervention, after the first 15, 30 and 45 min of the intervention and post-intervention.	All inpatients participated in the 3 different interventions. They were group 60 min sessions, developed in different days. Sedentary control conditions: sitting with reading materials and board games. Outdoor condition: Nordic walking through a green natural environment. Indoor conditions: cycling in the gym inside the hospital.	Green natural environment	Yes (Sedentary control condition groups)	FS MSS	FAS
Korpela et al., 2016 (Finland) [28]	13	Quantitative Quasi-experimental non-randomized study	Pre-intervention, post-intervention, 3-month follow-up	8-week program with 2hr sessions once a week about psychoeducation. Every second meeting was held at a preselected place in a green environment, and every second time indoors at the psychiatric clinic. Indoor sessions included psychoeducation focusing on depression, and outdoor sessions included psychological tasks that use the benefits of natural environments in addition to walking.	Park or Urban woodland	No	BDI-II	ROS SWEMWBS
Szczepanska-Gieracha et al., 2021 (Poland) [29]	25	Quantitative Parallel-group RCT	Pre-intervention, post-intervention and 2-week follow-up	Both groups (intervention and control) participated in a psychoeducational program and general fitness training. In addition, the intervention group completed 8 sessions (twice a week) of virtual reality	Virtual Reality with total immersion in nature	Yes (Psychoeducation and general fitness training)	GDS-30	HADS

Abbreviations. BDI-II: Beck Depression Inventory. FAS: Felt Arousal Scale. FS: Feeling Scale. GDS-30: Geriatric Depression Scale. HADS: Hospital Anxiety and Depression Symptoms. HRSD: Hamilton Rating Scale for Depression. MADRS: Montgomery–Asberg Depression Rating Scale. MSS: Mood Survey Scale. PANAS: Positive and Negative Affect Schedule. ROS: Restorative Outcome Scale. SF-36: Short Form Health Survey Questionnnaire. SWEMWBS: Short Warwick-Edinburgh Mental Well-Being Scale.

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
