# Peer review of "Lifestyle Prescription for Depression with a Focus on Nature Exposure and Screen Time: A Narrative Review"

_ijerph, 2022, doi:10.3390/ijerph19095094_

Round 1
Reviewer 1 Report
The overview examined the role of nature exposure and screen time in depression.with a focus on clinical samples on patients with well-defined depression.
The manuscript is scientifically valid and is clear, the references cited are current.Present several limitations due to the heterogeneity of the studies and methods used. The conclusions are consistent with the arguments described.
Reviewer 2 Report
This is a highly relevant review narratively summarizing evidence on the association between nature exposure, screen time and depressive symptoms/depression. While evidence for screen time is inconsistent, there is a consensus that natural environments/nature-based interventions improve depressive symptoms. Both nature and screen time exposure are highly changed during times of the COVID-19 pandemic and are still underrepresented in randomized treatment studies which are highly needed for the development of concrete treatment suggestions. Thus, the review covers a highly relevant topic but it lacks methodological clarity in its present form.
It would be preferable to conduct a systematic literature research. In contrast to the statement of the authors, a systematic review does not exclude a state-of-the-art description of the subject. Reliance on guidelines for systematic reviews like the PRISMA items secure an unbiased and complete inclusion of relevant literature, which is replicable by other authors. Especially for screen time, just using the search terms “screen time” and “media use” does not seem comprehensive.
In the introduction, NBIs are defined as ‘planned, intentional activities to promote individuals’ optimal functioning, health and well-being or to enable restoration and recovery through exposure to or interaction with either authentic or technological nature’, are there no studies using technological nature such as virtual reality etc?
In the introduction, the authors state that they prioritized studies with clinical samples composed of patients diagnosed with standardized criteria of depression, however, the review includes rather few studies with clinical depression patients but rather mixed samples or participants with depressive symptoms. The authors should clarify whether they want to focus on patients with clinical depression or adopt a broader focus.
Regarding results, the papers reviewed need to be described in more detail using a consistent strategy including which participants were included, how many, concrete effect sizes and outcome measures etc. Further, the structure of the manuscript needs to be clarified, it seems as if for the observational studies there are no studies with clinical depression patients and for the experimental studies it seems that they are not really experimental, they should better be renamed such as intervention studies … For these studies the authors first review studies with mixed patient samples followed by clinical depression samples, however the latter are not included in the first references of experimental studies.
Why does Table 1 only include patients with clinical depression and there is no table including the other studies? Additionally, the column headings in Table 1 should be clear, e.g., “evaluations” does not seem to cover which evaluations were performed but rather their timing. Also control groups should always be described in detail. In Table 1, what does 15 min s mean?
What is the advantage of simultaneously reviewing nature and screen time exposition, when there are no studies using both exposures and no theoretical frameworks how these exposures are linked? In the discussion, one study is named (reference 11), which is however not described in more detail. Regarding the interactive perspective, screen time and nature exposure are related because if subjects have a lot of sedentary screen time they have less time for PA in natural environments. It also has to be regarded, that subjects can also look at smartphone screens when they are in natural environments or that they can use screens to experience natural environments in nature documentations etc. Screens are now also often used to support physical activity with online sports courses. The effects of screen time on mental health might not only differ according to screen type but also be influenced by further patient characteristics, which needs to be discussed further.
The paper needs to be proofread by a native English speaker because there are a lot of grammar and spelling mistakes.
Reviewer 3 Report
Dear authors, following are suggestions to improve your manuscript.
It is not clear why a very classical narrative review was performed, because today there are numerous different types of reviews (scoping review, integrative review,….) with guidelines to perform and present the work. I suggest to choose a specific review type and to revise the review.
For the result section I would suggest to structure the literature not to the types of study design. It would be better to organize the structure according to the specific content you found for nature and screen time. At the moment your manuscript also lacks subtitles for your topic screen time.
Your conclusion and abstract sections are very general and unspecific. I suggest to show the reader the main content you found according to your topic’s nature (e.g. exercising in nature and nature based interventions are very unspecific for the reader) and screen time.
Round 2
Reviewer 3 Report
Dear author, the abstract and conclusion is now better than in the last version, but I miss a better research design and structur of the results.
Due to the large number of reviews nowadays, it would have been possible to use a different review type with guideline; the structure of the results does not fit very well with the aims in the introduction and should be revised.
The aim(s) in the abstract and the introduction differ (Abstract: We aimed to gather observational and intervention studies involving nature exposure and screen time. Introduction: The aim of this review was to gather the literature regarding the role of nature exposure and screen time in depression, with an emphasis on clinical samples of patients with well-defined depression. A second goal was to suggest an agenda for clinical practice and research).
In the response letter you say: one of the aims of the present review is to emphasize the different methodological approaches that have been used in the field. Thus, we finally decided to organize the presentation of the studies based on study design. This aim is not clearly stated in the abstract and introduction.
